# Psychometric Evaluation of the Korean Version of the Perceived Costs and Benefits Scale for Sexual Intercourse

**DOI:** 10.3390/healthcare11152166

**Published:** 2023-07-30

**Authors:** Hee-Jung Jang, Jungmin Lee, Soo-Hyun Nam

**Affiliations:** 1School of Nursing, Hallym University, Chuncheon-si 24252, Gangwon-do, Republic of Korea; hjjang@hallym.ac.kr; 2Department of Nursing, Andong National University, Andong 36729, Gyeongsangbuk-do, Republic of Korea; snam@anu.ac.kr

**Keywords:** condom use, safe sex, schools, sexual behavior, social norms

## Abstract

This study aimed to evaluate the psychometric properties of the Korean version of the perceived costs and benefits scale (K-PCBS) for sexual intercourse to deepen our understanding of the complex psychosocial and interpersonal elements influencing young people’s sexual decision-making. The study participants were 227 college students in South Korea aged 18–25 years. Two factors were extracted through factor analysis, accounting for 61.05% and 58.18% of the variance for perceived costs and perceived benefits, respectively, and showed a good model fit in the K-PCBS. Reliability was high, as indicated by Cronbach’s alphas of 0.87 and 0.84 for the perceived costs and perceived benefits subscales, respectively. The results indicate that the K-PCBS may serve as an appropriate instrument for measuring South Korean youth’s cost and benefit analysis regarding their sexual decision-making. Our study makes a significant contribution to the literature and field because it identifies the underlying feelings and attitudes of young adults toward engaging in sexual activities using the K-PCBS for sexual intercourse, which has good reliability, validity, and internal consistency.

## 1. Introduction

Early adulthood is a period in which sexual curiosity and sexual drive increase according to physical development, and sexual stimulation is actively pursued [1]. Unlike in previous generations, today’s reality of an increasingly open sex culture means that young people’s dating or sexual experiences cannot be viewed as unconditionally problematic or negative and be suppressed [2]. However, due to the cultural characteristics of South Korea, it may be difficult for youths to obtain sexual information freely and openly, which may increase their likelihood of engaging in risky sexual behavior [3]. Sexual activity among young people is often considered problematic because of its potential risks to health and life prospects, and its negative consequences for wider society. Sexual behaviors that have risk factors that can negatively affect individuals’ physical and mental health [4], such as premarital pregnancy, sexually transmitted infections, unwanted sex, and sexual violence, have both short- and long-term effects on young people; therefore, more attention should be paid to their prevention and treatment.

Young people perceive and imitate behaviors observed at home, at school, and in their peer groups during their developmental process. While similar to their adolescent development, young adults do not unconditionally accept and follow the actions of others, but rather, choose actions according to their own judgment ability, forming their own unique behaviors and thinking systems. Furthermore, when making a judgment during young adulthood, in which psychosocial development through social and cultural factors is still in progress, their choices and actions are informed by various events, experiences, and changes that they constantly experience in their social environment [5,6,7,8]. Therefore, it is crucial to highlight the need to increase our understanding of complex psychosocial and interpersonal elements influencing young people’s sexual decision-making [9].

In this regard, it is important to understand the meaning of problematic or risk-taking behavior in young people. Assessing the youths’ beliefs and thoughts about what influences them to behave in potentially health-compromising ways can play a key role in understanding their behavior. However, in South Korea, limited tools exist to measure the costs and benefits that young people assess in the process of sexual decision-making. There is a need for measurement tools that focus on the view that understanding the positive and negative consequences associated with young people’s decision-making is important in understanding why they become sexually active. The perceived costs and benefits scale (PCBS) of sexual intercourse was developed to measure adolescents’ perceived costs and benefits of non-marital sexual intercourse. It emphasizes the importance of understanding the costs and benefits of sexual intercourse from the youth’s perspective, as their perceptions determine their choices. Therefore, this study aims to adapt the PCBS cross-culturally for sexual intercourse into the Korean context by evaluating the psychometric properties of the Korean version of the scale (K-PCBS).

## 2. Methods

### 2.1. Design

This methodological study examined the psychometric properties of the K-PCBS for college students in South Korea.

### 2.2. Participants

Convenience sampling was employed to recruit respondents, primarily through an online survey company (i.e., Macromill Embrain). The participants in this study were college students aged 18–25 years. The inclusion criteria were as follows: (1) being aged 18–25 years old; (2) has engaged in sexual intercourse in the past (or is currently sexually active); (3) has Internet access. An information sheet was posted on the front page of the survey, detailing basic eligibility requirements and instructing participants to click a link if they were interested in participating.

The appropriate sample size for a tool development study is five to ten times the number of items in the tool [10,11]. The PCBS has two subscales (the perceived costs and perceived benefits subscales), each comprising 10 items. Therefore, approximately 200 participants (100 participants for each subscale) were required for this study [12]. Considering a 15% incomplete and anticipated drop-out rate, 230 participants were recruited. The final sample size was 227 owing to incomplete responses and outliers (119 students for perceived costs and 108 students for perceived benefits).

## 3. Measurements

### 3.1. Perceived Costs and Benefits Scale (PCBS) for Sexual Intercourse

In the original mixed methods study, the scale was developed over many years and included extensive interviews with a diverse sample of youths [13]. In a sample of over 2400 students in grades 7–12, multiple hypotheses regarding gender, behavioral status, and grade differences were investigated [13]. The PCBS was developed to measure the costs and benefits assessed by the youth when deciding whether to have sex. Current measurements assume that viewing the youth as decision-makers and understanding the positive and negative consequences associated with them engaging in sexual behaviors is important in understanding why they are sexually active. The scale consists of two independent subscales comprising 10 items each, regarding the perceived costs and benefits of engaging in or not engaging in sexual intercourse. Depending on their sexual experiences, participants answer different questionnaires. Responses are scored on a 4-point Likert scale ranging from 0 (strongly disagree) to 3 (strongly agree). The Cronbach’s alphas for perceived costs and perceived benefits were 0.70 and 0.65, respectively, in the original study [13], and 0.87 and 0.84, respectively, in this study. Permission to use this tool was obtained from the original author.

While acknowledging the merits of alternative measures for assessing reliability in ordinal variables, we opted to utilize Cronbach’s alpha to allow for consistency in the evaluation of the scale’s reliability and to facilitate meaningful comparisons with existing research findings in sexual decision-making studies. Future studies can explore the use of alternative measures, such as McDonald’s omega or ordinal alpha, to further validate and enhance the assessment of reliability for Likert-type scales in the specific context of sexual decision-making among South Korean youths. Comparing and contrasting different measures of reliability could provide valuable insights and contribute to refining the measurement practices in this field.

### 3.2. Translation Procedures

The following procedures were based on the 8-step tool development and verification process suggested by DeVellis [12]. In this study, since the existing English tool was translated into Korean, the initial three steps were tool component confirmation, item writing, and tool scale determination. Approval for use and replacement by translation and back-translation processes was obtained [14].

First, the researcher sought approval to use the tools from Small [13], the developer of PCBS for sexual intercourse, via e-mail. Subsequently, two independent translators, whose native language was Korean, translated the scale from English into Korean. A third independent translator, fluent in both English and Korean, then performed a reverse translation. Both versions were reviewed by the first and second authors to identify and compare discrepancies between the original English version and the translated Korean version and to ensure that the meaning of each item was retained. None of the three translators had majored in nursing; therefore, after receiving feedback from four nursing professors on the appropriateness of the selected vocabulary, clarity of translation, and whether any items needed to be corrected due to cultural differences, the items were modified within the scope of retaining the significance of the existing items. Finally, five college students who participated in a facial validity verification test were excluded from this survey, and the final version of the cross-culturally adapted K-PCBS tool was not modified.

### 3.3. Data Analysis

For data analysis, the content validity, construct validity, criterion validity, and reliability were verified based on DeVellis’s tool development guidelines. The content validity index (CVI) was used to verify the content validity, and exploratory factor analysis (EFA) and confirmatory factor analysis (CFA) were performed to confirm construct validity. The reliability test examined internal consistency to check whether the items consistently measured the construct and if Cronbach’s alpha was confirmed. In addition, the degree of sexual communication with parents and peers according to participants’ sexual experiences was analyzed using descriptive statistics. Group differences in participants’ test scores according to participant characteristics were analyzed using a *t*-test, one-way ANOVA, and Scheffe’s test. All analyses were conducted using SPSS 26.0 statistical software.

### 3.4. Ethical Considerations

Approval to conduct the research was obtained from the Institutional Review Board of Hallym University (HIRB-2021-075). For ethical protection of the research participants, prior to data collection, the purpose and contents of the research were explained to them, and they were informed that they could withdraw from the study at any time. Written informed consent was then obtained from those who agreed to participate in the research after they were assured that their responses would be anonymous and would not be used for any purpose other than this research study.

## 4. Results

### 4.1. General Characteristics of Participants

Participants’ general characteristics are illustrated in Table 1. The study participants were 227 Korean college students aged 18–25 years. We divided the participants into two groups based on their experiences of sexual intercourse to explore the differences in the decision-making influences on their sexual behavior. Participants with no past sexual experience had a mean age of 21.65 ± 1.898 years. College grade level was equally divided into freshman, sophomore, junior, and senior, with the number of participants in each grade level ranging from to 24–36 (20.2% to 30.3%). Most of the students lived with their families. Only one out of 10 students were currently dating, with the average duration of their relationship being around 14.11 ± 18.977 months, while approximately 60% had not been in a relationship previously (*n* = 68). Nearly 60% of students answered that they had moderate sexual tolerance, which refers to an individual’s acceptance, open-mindedness, and non-judgmental attitude towards diverse sexual orientations, identities, behaviors, and practices. Sexual tolerance reflects an individual’s ability to respect and recognize the rights and choices of others regarding their sexual lives, regardless of whether those choices align with their own personal beliefs or preferences. The methods used when dealing with sexual concerns or problems and when obtaining sex-related information mostly involved the use of the Internet, such as blogs or social media, and consulting with peers and seniors (84% (*n* = 100) and 90% (*n* = 107), respectively).

The average age of participants who already had sexual experience was 22.67 ± 1.635 years. Seven out of 10 students were either juniors or seniors and living with their families. Among them, more than half were currently dating, and the mean duration of their relationships was 20.73 ± 15.159 months. Similar to students without sexual experience, the group who already had sexual experience indicated that they had moderate sexual tolerance. More than 90% of the students used the Internet or consulted their friends or seniors when dealing with sexual concerns or when they had problems obtaining sexual information (*n* = 97 and *n* = 101, respectively).

### 4.2. Descriptive Statistics and Mean Scores of the K-PCBS

The normality of the distributions was supported by the skewness and kurtosis coefficients (Table 2). Non-parametric tests are robust against violations of normality assumptions and provide reliable results even when the data deviates from a normal distribution [15]. However, departures from normality can still affect the power and interpretation of these tests. Therefore, it is essential to evaluate the distribution of the data and assess any significant departures from normality. Hence, although an ordinal qualitative variable does not strictly adhere to a normal distribution, it is still important to acknowledge and address departures from normality when applying non-parametric tests. This approach ensures the robustness and reliability of the statistical analysis conducted on the dataset.

For those who have had no sexual experience in the past, the mean score on the K-PCBS was 2.14 ± 0.617 points. Here, item PC-8, “I do not have sex because I have not met someone I truly love” had the highest score (3.09 ± 0.911), followed by item PC-10, “I do not have sex because my or my partner’s unwanted pregnancy could ruin my future life” (2.55 ± 0.946). The items that scored the lowest were “I do not have sex because my friends around me do not agree to have sex with me” (1.55 ± 0.767) and “I do not have sex because my parents do not allow it” (1.67 ± 0.845).

For students who had engaged in sexual intercourse, the mean score on the K-PCBS was 1.91 ± 0.505 points. Item PB-5, “I have sex because it makes me feel good” had the highest score (2.81 ± 0.767), followed by item PB-6, “I have sex because it makes me feel loved” (2.69 ± 0.791). The items that showed the lowest scores were item PB-3, “I have sex because I want to become pregnant or become a parent” and item PB-10, “I have sex because I think it is a wonderful thing that people I admire do” (1.42 ± 0.725 and 1.46 ± 0.633, respectively).

### 4.3. Validity Test for the K-PCBS

#### Content Validity

The mean item–content validity index (I-CVI) of 1.0 was computed by four experts in adult nursing and child and adolescent nursing, suggesting a 100% content agreement on all items of the K-PCBS [10]. The experts who participated in the evaluation judged that the K-PCBS was appropriate in terms of social and cultural aspects and explained that it was consistent with the evaluation of the existing PCBS. The terms chosen by the experts have been slightly altered to better fit the target population.

### 4.4. Construct Validity

#### EFA with Varimax Rotation

The results of the EFA with varimax rotation are illustrated in Table 3. To confirm whether the data of this study were suitable for factor analysis, the Kaiser–Meyer–Olkin (KMO) sample fit measure and Bartlett’s sphericity verification were performed. If the KMO value was 0.50 or more, factor analysis could be performed, while a value greater than 0.80 indicated a good model fit. Additionally, Bartlett’s sphericity test indicated that the sample size was appropriate for the number of questions when the *p*-value was <0.05. In this study, KMO was 0.858 for perceived costs and 0.839 for perceived benefits, indicating that the selection of variables for factor analysis was appropriate and that Bartlett’s sphericity test had a model suitable for factor analysis (χ^2^ = 519.228, *df* = 45, *p* < 0.001 and χ^2^ = 387.423, *df* = 45, *p* < 0.001, respectively).

Two factors were extracted for each subscale, accounting for 61.045% and 58.177% of the variance of each of the 10 items in the perceived costs and perceived benefits scales, respectively. The factor loadings of all the items were above 0.30. The scree plot, depicted in Figure 1, shows a sharp decline in the slope for the two factors in each sub-category. In perceived costs, Factor 1 was titled “social and cultural norms and barriers” and Factor 2 was titled “worried about negative consequences”. Both factors consisted of five items each. Regarding perceived benefits, Factor 1 was titled “making social relationships” and contained six items, and Factor 2 was titled “learning more about myself” and contained four items.

### 4.5. Measuring Model Using CFA

CFA was performed on the two factors extracted using EFA. The results for both the perceived costs and perceived benefits showed that the factor loadings of all the items were above 0.30 (Figure 2). Measuring the model fit for perceived costs showed a good model fit (χ^2^ = 71.653 (*df* = 34, *p* < 0.001), CMIN/DF = 2.11, RMSEA = 0.097 (90%CI: 0.07–0.13), and CFI = 0.92). For perceived benefits, the model also showed a good model fit (χ^2^ = 69.396 (*df* = 34, *p* < 0.001), CMIN/DF = 2.04, RMSEA = 0.099 (90%CI: 0.07–0.13), and CFI = 0.90). Since the null hypothesis of the chi-squared test is that the model-implied covariance matrix is exactly equal to the observed covariance matrix in the population, the *p*-value should be not significant (*p* > 0.05) to show a good model fit. Except for the *p*-value, the results indicated that the comprehensive model is suitable and adequately describes our model by meeting all recommended level criteria (Table 4).

### 4.6. Reliability for the K-PCBS

The Cronbach’s alpha values of the K-PCBS were 0.867 and 0.838 for perceived costs and perceived benefits, respectively. The reliability values for each factor were 0.842 for “social and cultural norms” (Factor 1 on perceived costs), 0.798 for “negative consequences” (Factor 2 on perceived costs), 0.843 for “social relationship” (Factor 1 on perceived benefits), and 0.685 for “looking back on myself” (Factor 2 on perceived benefits). These results were considered to have high-to-moderate reliability.

### 4.7. Descriptive Statistics of the K-PCBS Using Two Sub-Categories

There were statistically significant differences between the perceived costs and perceived benefits scores of the male and female participants (Table 5). Among the female participants, perceived costs were significantly higher and perceived benefits were significantly lower (t = 2.725, *p* = 0.007; t = 3.421, *p* = 0.001, respectively). In addition, participants’ sexual tolerance showed a statistically significant association with perceived costs (F = 5.116, *p* = 0.007). Sexual tolerance was significantly higher among students with conservative attitudes than among those who had moderate (*p* = 0.03) or open (*p* = 0.02) attitudes. However, the Scheffe post-hoc test showed no significant difference in perceived benefits within the sub-categories (*p* = 0.611).

## 5. Discussion

This study translated the PCBS into Korean for a cross-cultural adaptation to the Korean youth and established the adapted scale’s psychometric properties. Social norms and interpersonal relationships among young adults and their peers and parents play significant roles in determining the actual sexual behavior of the youth; therefore, it is essential to understand the context of the influence these factors have on the youth’s perception of sexual behavior. Several different evaluations of the psychometric properties, including content validity, construct validity, and reliability, were verified using 227 college student participants in South Korea. These evaluations may contribute to the development of valid tools for measuring the youth’s sexual decision-making and perceptions, which may influence their actual sexual behaviors. The results revealed that the K-PCBS showed acceptable internal consistency (Cronbach’s α = 0.867 for the perceived costs scale and 0.838 for the perceived benefits scale) and validity (I-CVI = 1.0).

In this study, the EFA with varimax rotation revealed that two factors were extracted for each subscale, each comprising 10 items. For perceived costs, Factor 1 was titled “social and cultural norms and barriers” and Factor 2 was titled “worried about negative consequences”. For Factor 1, the items with the highest score were “I do not have sex because having sex does not make me happy” and “I do not have sex because I feel guilty about having sex”. This result is in line with other findings in the Korean sociocultural context regarding guilty feelings or negative perceptions about sexual intercourse during early adulthood [16]. Owing to the influence of the traditional Confucian culture, the topic of sexuality is considered taboo in South Korea [16], which discourages the youth from developing liberal and active attitudes toward sexual activity. Such sociocultural influences hinder Korean college students from obtaining the necessary information related to sex; furthermore, systematic sex education is rarely provided to them [17]. As the perception of sex-openness is often reported to be different between Korean and Western cultures, the findings of this study must be compared further with those of other studies conducted in different settings and cultures. Previous studies have found a significant difference between the rate of sexual experience between Korean (4.3%) and American youths (46.8%) [18]. Additionally, when advocating and disseminating sex education, it is necessary to consider the various sociocultural factors at play, such as the individual’s background, area of residence, race, and ethnicity.

The items with the highest score among Factor 2, titled “worried about negative consequences,” were “I do not have sex because I have not met someone I truly love” and “I do not have sex because my or my partner’s unwanted pregnancy could ruin my future life”. These results may reflect the youth’s fear of unwanted pregnancies adversely affecting their futures. Unplanned pregnancy and sexually transmitted diseases are two major reasons young people avoid having sex [19,20]. According to previous research, 54.6% of Korean college student participants answered that they do not use contraception, and only 31.9% answered that they always do [21]. The low use of contraception among Korean youths is the greatest contributor of the increase in unwanted pregnancies or abortions and the rise in the risk of sexually transmitted infections [22,23]. Additionally, the World Health Organization has reported that the youth are unaware of how to avoid pregnancy or gain access to birth control pills, including emergency contraceptives [24]. Therefore, it is critical to develop and improve the youth’s access to practical education on sexual health, including contraception methods, prevention of and protection against sexually transmitted diseases, and support systems and services for those dealing with unplanned pregnancies.

Regarding perceived benefits, Factor 1 was titled “making a social relationship” and Factor 2 was titled “knowing more about myself”. In Factor 1, the item with the highest score was “I have sex because it makes me feel like an adult”. This result is consistent with that in previous studies that showed that the youth experience sexual behavior as a meaningful social act [25]. The youth experience multiple aspects of life changes during this critical transition period into adulthood such as independence from parents, developing meaningful relationships with partners, and higher educational achievement. Above all, sexual experience, embedded with social and relational significance, is considered as one of the most important factors in “the transition to adulthood [26]”. Moreover, as socialization with peers and partners is an important social referent for engaging in either risky or safe sexual behavior during this period [7,11], peer or partner education could be an effective strategy for establishing safe sexual values and sexual health statuses among the youth.

In Factor 2, which was titled “knowing more about myself,” the items with the highest score were “I have sex because it makes me feel good” and “I have sex because it makes me feel loved”. Previous studies have stated that sexual behavior improves the youth’s self-esteem and makes them experience a state of positive mental health [25]. Based on the results of this study, it is noteworthy that, as they mature, the youth tend to develop a better understanding of themselves and become more aware of their feelings.

To the best our knowledge, this is the first study to verify the reliability and validity of the Korean version of the PCBS. It is difficult to directly compare the findings of this study with those of previous research as none have confirmed the validity and reliability of the PCBS, except for those conducted on the original tool. However, the adequate internal consistency and validity in this study showed that the K-PCBS is acceptable and useful. Additionally, the K-PCBS consists of 20 items with an advantage of being less time-consuming, making it a practical tool to measure the youth’s perception of the costs and benefits of engaging or not engaging in sexual activities. Health providers can utilize this tool to deepen their understanding of the psychosocial context influencing youth’s sexual decision-making.

There are certain limitations to this study. Firstly, test–retest reliability was not performed, necessitating its further evaluation, as well as that of other multiple psychometric properties, in future studies. Secondly, the findings cannot be generalized to other populations as the study only recruited college students in South Korea as participants. Furthermore, the study did not specifically consider the intentional or situational aspects of students’ sexual behavior, which could impact the interpretation of the perceived costs and benefits measured by the K-PCBS. Sexual behavior is influenced by personal intentions, circumstances, and situational factors, and the lack of differentiation in this study limits understanding of the nuanced interplay between perceived costs and benefits. Respondents’ answers may have been influenced by situational factors or external constraints, which were not explicitly accounted for in the analysis.

Considering the sociocultural context of Korea where sexual intercourse among the youth is restricted, it is important to recognize the potential discrepancy between reported behavior and actual behavior due to social desirability bias and cultural factors. There is a possibility of bias within this sociocultural context, and respondents may have engaged in sexual intercourse despite reporting otherwise. This introduces uncertainty in interpreting the findings. Additionally, the tool used in this study measures the benefits and costs of having sex, which may further contribute to social desirability bias and influence the responses. It is therefore crucial to acknowledge the limitations of our knowledge regarding how the youth behave in real-life situations.

To address these limitations and gain a comprehensive understanding of the perceived costs and benefits of sexual decision-making among Korean youths, future studies could incorporate methods that capture the intentional and situational aspects of participants’ sexual behavior. Qualitative approaches, contextual information, and retrospective assessments could provide deeper insights into decision-making complexities and refine the measurement of costs and benefits in this domain.

Despite these limitations, the findings of this study contribute preliminary evidence supporting the acceptable reliability and validity of the K-PCBS when applied to young people in the Korean context. This underscores the need for further research that addresses the identified limitations, aiming to enhance understanding of youth sexual behavior decision-making and promote more accurate and nuanced assessments in future studies.

## 6. Conclusions

This study was conducted to verify the reliability and validity of the K-PCBS for college students in South Korea. From the results, it can be concluded that K-PCBS has acceptable psychometric properties. This study not only promotes a deeper understanding of the psychosocial and interpersonal factors that influence youth’s sexual decision-making, but also contributes to measuring the costs and benefits of perceived sexual behavior among the Korean youth. By evaluating the consequences and benefits of engaging in sexual intercourse and understanding the sexual activity of the youth using this scale, researchers will be more informed about how to respond to youths regarding at-risk sexual behavior in a more sensitive manner in both clinical and community settings. This scale can also be useful to health workers planning to provide adequate sexual education interventions to the youth.

The validation of the K-PCBS for the Korean population not only allows for a more comprehensive and culturally sensitive exploration and measurement of the costs and benefits associated with sexual decision-making, but also aids in better understanding the factors that shape the sexual behaviors and choices of young Koreans. This fills a significant gap in the literature and paves the way for future research and interventions aimed at promoting healthy decision-making in this context. Further studies should investigate the scale’s cross-cultural validity and encompass more diverse samples to enhance the generalizability of the findings. Overall, the introduction and validation of the K-PCBS as a novel tool for measuring the costs and benefits of sexual decision-making among young Koreans provides valuable insights into the psychosocial and interpersonal elements that influence this domain.

## Figures and Tables

**Figure 1 healthcare-11-02166-f001:**
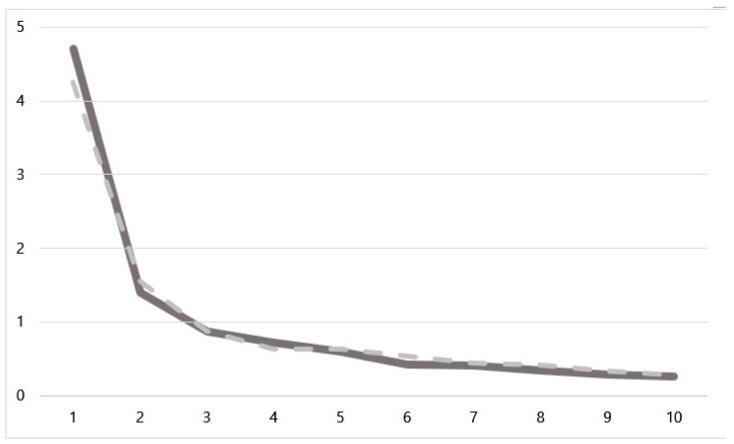
Scree plots for each category. Solid line: Perceived Cost (*n* = 119) who have not had a sexual experience; Dotted line: Perceived benefits (*n* = 108) who had a sexual experience.

**Figure 2 healthcare-11-02166-f002:**
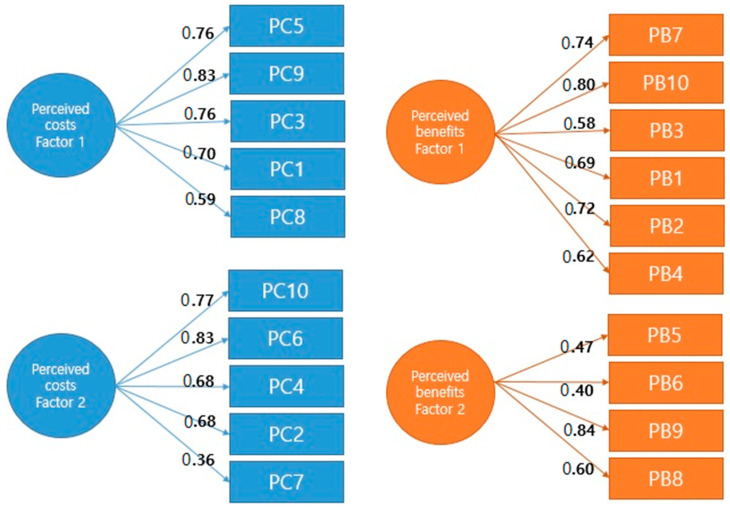
Path diagram with standardized estimates.

**Table 1 healthcare-11-02166-t001:** General characteristics of participants based on their sexual intercourse experience (*N* = 227).

	Category	Have Not Had Sexual Intercourse (*n* = 119)			Have Had Sexual Intercourse (*n* = 108)		
		*n*	(%)	Mean ± SD (Min–Max)	*n*	(%)	Mean ± SD (Min–Max)
Gender	Female	68	57.1		58	53.7	
	Male	51	42.9		50	46.3	
Age				21.65 ± 1.898 (19–25)			22.67 ± 1.635 (19–25)
Grade level	Freshman	24	20.2		10	9.3	
	Sophomore	36	30.3		17	15.7	
	Junior	27	22.7		41	38.0	
	Senior	32	26.9		40	37.0	
Current residential status	Alone	13	10.9		23	21.3	
	Family (Parents, brothers/sisters, or relatives)	96	80.7		74	68.5	
	Apartment with roommate (Same-sex and opposite sex)	1	0.8		9	8.3	
	Dormitory	9	7.6		2	1.9	
Currently dating	Yes (Month)	9	7.6	14.11 ± 18.977 (1–56)	60	55.6	20.73 ± 15.159 (2–61)
	No but in the past	42	35.3		43	39.8	
	Never	68	57.1		5	4.6	
Method used to deal with sexual concerns or problems	Friend or senior	42	35.3		61	56.5	
	Parents or family	14	11.8		6	5.6	
	Internet, such as blogs or social media	58	48.7		36	33.3	
	Others (Alone, partner, hospital, seminars, etc.)	5	4.2		5	4.6	
Sexual tolerance	Conservative	33	27.7		15	13.9	
	Moderate	71	59.7		66	61.1	
	Open	15	12.6		27	25.0	
Siblings	No siblings (Only child)	9	7.6		9	8.3	
	Sister (Sister, younger sister)	45	37.8		38	35.2	
	Brother (Brother, younger brother)	55	46.2		50	46.3	
	Sisters and brothers	10	8.4		11	10.2	
Most used method to obtain sex-related information	Friend or senior	23	19.3		33	30.6	
	Parents or family	2	1.7		4	3.7	
	Internet, such as blogs or social media	84	70.7		68	62.9	
	Others (Alone, book, seminars, etc.)	10	8.3		3	2.8	

**Table 2 healthcare-11-02166-t002:** Korean Perceived Costs and Benefits Scale for Sexual Intercourse (K-PCBS) (*N* = 227).

No.	Items	Mean ± SD	Skewness	Kurtosis
Perceived cost (*n* = 119) for those who have not had a sexual experience				
PC-1	I do not have sex because I think it is morally wrong or against my religious beliefs.	1.69 ± 0.841	1.165	0.796
PC-2	I do not have sex because it puts me at risk of contracting a sexually transmitted disease or AIDS.	2.20 ± 0.935	0.091	−1.075
PC-3	I do not have sex because my parents do not allow it.	1.67 ± 0.845	1.201	0.834
PC-4	I do not have sex because I do not consider myself mature enough to do so.	2.38 ± 1.066	0.090	−1.234
PC-5	I do not have sex because my friends around me do not agree to have sex with me.	1.55 ± 0.767	1.419	1.717
PC-6	I do not have sex because me or my partner may become pregnant.	2.40 ± 1.003	−0.012	−1.091
PC-7	I do not have sex because I have not met someone I truly love.	3.09 ± 0.911	−0.937	0.238
PC-8	I do not have sex because having sex does not make me happy.	2.15 ± 0.945	0.366	−0.795
PC-9	I do not have sex because I feel guilty about having sex.	1.73 ± 0.851	1.053	0.484
PC-10	I do not have sex because my or my partner’s unwanted pregnancy could ruin my future life.	2.55 ± 0.946	−0.074	−0.875
Total		2.14 ± 0.617		
Perceived benefits (*n* = 108) for those who had a sexual experience				
PB-1	I have sex because it helps me forget the problems I am facing.	1.70 ± 0.800	0.701	−0.729
PB-2	I have sex because it makes me feel like an adult.	1.74 ± 0.802	0.617	−0.826
PB-3	I have sex because I want to become pregnant or become a parent.	1.42 ± 0.725	1.721	2.292
PB-4	I have sex to have or make friends of the opposite sex.	1.64 ± 0.791	0.974	0.045
PB-5	I have sex because it makes me feel good.	2.81 ± 0.767	−1.045	1.090
PB-6	I have sex because it makes me feel loved.	2.69 ± 0.791	−0.666	0.157
PB-7	I have sex because I want to hang out with my friends.	1.48 ± 0.755	1.592	2.059
PB-8	I have sex because I want to know what it feels like to have sex.	2.17 ± 0.942	0.000	−1.304
PB-9	I have sex because it makes me feel confident about myself.	1.96 ± 0.875	0.329	−1.046
PB-10	I have sex because I think it is a wonderful thing that people I admire do.	1.46 ± 0.633	1.043	0.027
Total		1.91 ± 0.505		

**Table 3 healthcare-11-02166-t003:** Results of the exploratory factor analysis (EFA).

Item		Factor	
		1	2
Perceived cost (*n* = 119) for those who have not had a sexual experience			
1 (PC-5)	I do not have sex because *my friends* around me do not agree to have sex with me.	0.776	
2 (PC-9)	I do not have sex because I *feel guilty* about having sex.	0.773	
3 (PC-3)	I do not have sex because *my parents* do not allow it.	0.740	
4 (PC-1)	I do not have sex because I think it is morally wrong or against my *religious beliefs*.	0.681	
5 (PC-8)	I do not have sex because having sex *does not make me happy*.	0.472	
6 (PC-10)	I do not have sex because my or my partner’s unwanted *pregnancy* could ruin my future life.		0.758
7 (PC-6)	I do not have sex because me or my partner may become *pregnant*.		0.729
8 (PC-4)	I do not have sex because I do not consider myself *mature* enough to do so.		0.590
9 (PC-2)	I do not have sex because it puts me at risk of contracting a *sexually transmitted disease or AIDS*.		0.548
10 (PC-7)	I do not have sex because I have not met someone I *truly love*.		0.431
Eigenvalue		4.706	1.399
Total variance explained proportion (%)		47.056	13.990
Cumulative proportion (%)		47.056	61.045
Perceived benefits (*n* = 108) for those who had a sexual experience			
11 (PB-7)	I have sex because I want to hang out with my *friends*.	0.805	
12 (PB-10)	I have sex because I think it is a *wonderful thing* that people I admire do.	0.765	
13 (PB-3)	I have sex because I want to become *pregnant* or become a parent.	0.644	
14 (PB-1)	I have sex because it *helps me forget* the problems I am facing.	0.629	
15 (PB-2)	I have sex because it makes me *feel like an adult*.	0.625	
16 (PB-4)	I have sex to have or *make friends* of the opposite sex.	0.515	
17 (PB-5)	I have sex because it makes me *feel good*.		0.709
18 (PB-6)	I have sex because it makes me *feel loved*.		0.615
19 (PB-9)	I have sex because it makes me *feel confident* about myself.		0.517
20 (PB-8)	I have sex because I *want to know* what it feels like to have sex.		0.417
Eigenvalue		4.256	1.562
Total variance explained proportion (%)		42.562	15.615
Cumulative proportion (%)		42.562	58.177

**Table 4 healthcare-11-02166-t004:** Results of the confirmatory factor analysis (CFA).

Model	χ^2^	CMIN/DF	RMSEA	CFI
Criteria		<3	≤08	>0.90
Results for Perceived Cost	71.653 (*df* = 34, *p* < 0.001)	2.107	0.097 (90% CI: 0.07–0.13)	0.924
Results for Perceived Benefits	69.396 (*df* = 34, *p* < 0.001)	2.041	0.099 (90% CI: 0.07–0.13)	0.901

**Table 5 healthcare-11-02166-t005:** Difference in scores according to participants’ general characteristics (*N* = 227).

Characteristics	Categories	Perceived Cost (*n* = 119) for Those Who Have Not Had a Sexual Experience		Perceived Benefits (*n* = 108) For Those Who Had a Sexual Experience	
		Mean ± SD	t or F(*p*)	Mean ± SD	t or F(*p*)
Gender	Female	2.27 ± 0.637	2.725 (0.007) **	1.76 ± 0.508	3.421 (0.001) **
	Male	1.97 ± 0.550		2.08 ± 0.448	
Sexual tolerance	Conservative	2.41 ± 0.805 ^a^	5.116 (0.007) **	1.79 ± 0.652	0.494 (0.611)
	Moderate	2.07 ± 0.491 ^b^		1.93 ± 04.68	
	Open	1.89 ± 0.515 ^b,c^		1.93 ± 0.511	

** *p* < 0.1; Scheffe post-hoc test a > b > c.

## Data Availability

The data that support the findings of this study are available from the corresponding author on special request.

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
