# Peer review of "Psychometric Evaluation of the Korean Version of the Perceived Costs and Benefits Scale for Sexual Intercourse"

_healthcare, 2023, doi:10.3390/healthcare11152166_

Round 1
Reviewer 1 Report
Dear Authors.
The revised manuscript evaluates the psychometric properties of the Korean version of the Perceived Costs and Benefits Scale for sexual intercourse.
It provides an appropriate overview of the research in the abstract, although its different sections should be specified as indicated by journal guidelines: Background, Methods, Results, and Conclusions.
The introduction of this manuscript is well elaborated and adequately addresses the existing issue on the research topic and the importance of validating this scale for the Korean population.
The stated objective is adequate to clarify the stated study problem.
The methodology allows the authors to adequately address the study problem to achieve the proposed objective by following all the necessary requirements for the adaptation and validation of this American questionnaire to the Korean population, and complying with the ethical standards required for human studies.
The results are clearly presented for ease of understanding, although I wonder why absolute values of skewness less than 2 and kurtosis less than 7 are considered as normal distribution when it is usual to consider a normal curve if the skewness and kurtosis are between ± 0.5.
The discussion provides a detailed and in-depth analysis of the results obtained and establishes relationships with previous studies on the subject of this research.
The conclusions are coherent with the results obtained, responding adequately to the proposed objective and to the currently existing scientific evidence.
The references are appropriate to address this topic of study, although they should be adapted to the journal's standards.
Kind regards.

Author Response
Manuscript ID: healthcare-2505762 Result: Major Revision
We would like to express our appreciation for your extremely thoughtful suggestions. Your feedback was extremely helpful to strengthen our manuscript. As you will see below, we have been able to revise and improve the paper as a result of your valuable feedback.

Overall, we have made changes throughout the paper that address the points you have made as shown below. After correcting the manuscript according to the reviewers’ and editors’ comments, we got this paper revised by an academic revision company again. The corrected parts have been marked with Red Font and indicated with page numbers in the table below for easy reference.
Thank you again for taking the time to share your constructive feedback.
Yours sincerely,
The authors

Reviewer 2 Report
line-66 - descriptive research - details?
line 77 - 2) was past activity only criteria or currently active were also included? Was it determined if the lack of sexual activity intentional or circumstantial?
line 152 - define sexual tolerance
line118-119 - unclear - what test, which final version
line 128 - is it what was meant by descriptive research above? (statistics)
line 261-262 - "sex does not make me happy" seems ambiguous and personal (than social/cultural). Perhaps the question could be more explicit about the reason for the unhappiness.
Author Response

(The authors gave the same response as above.)

Round 2
Reviewer 1 Report
Dear Authors.
I still think that the statement "When the absolute value of skewness exceeds 2, items with an absolute value of 7 or more of kurtosis are out of normality" (lines 185-186). In the bibliographic reference on which you rely it says "Distribution 1 was multivariate normal with univariate skewness and kurtoses equal to 0. Distribution 2 was moderately nonnormal with univariate skewness of 2.0 and kurtoses of 7.0. Finally, Distribution 3 was severely nonnormal with univariate skewness of 3.0 and kurtoses of 21.0". I reaffirm after this and after an exhaustive search, that for it to be a normal distribution the values of skewness and kurtosis must be between ± 0.5, although I do not consider this to be of major importance, nor does it detract from the value of your research, because an ordinal qualitative variable does not have to follow a normal distribution, but this should be taken into account when using non-parametric tests in your analysis. Therefore, this statement should be corrected.
In Table 4 they state that the RMSEA (Root mean squared error of approximation) must have a value greater than 0.05 and this statement is erroneous because the RMSEA must be equal to or less than 0.05 for the model to fit the sample adequately. Although some authors consider RMSEA values less than or equal to 0.08 as an optimal fit. As this result is within the confidence interval the fit is adequate, although they should change it in the table and put less than 0.05 instead of greater.
Some references still do not conform to the journal's standards. For example, you have put: Finer LB, Phibin JM. Sexual initiation, contraceptive use, and pregnancy among young adolescents. Pediatrics. 2013;131(5),886-891. https://doi.org/10.1542/peds.2012-3495. And they should put it this way: Finer, L.B; Phibin, J.M. Sexual initiation, contraceptive use, and pregnancy among young adolescents. Pediatrics 2013,13,886-891. Therefore, they should be modified.
Kind regards.
Author Response
We would like to express our appreciation for your extremely thoughtful suggestions. Your feedback was extremely helpful to strengthen our manuscript. As you will see below, we have been able to revise and improve the paper as a result of your valuable feedback.

Overall, we have made changes throughout the paper that address the points you have made as shown below. After correcting the manuscript according to the reviewers’ and editors’ comments, we got this paper revised by an academic revision company again. The corrected parts have been marked with Red Font and indicated with page numbers in the table below for easy reference.
Thank you again for taking the time to share your constructive feedback.
Yours sincerely,
The authors
